# EMERDIFF: EMERGING PIXEL-LEVEL SEMANTIC KNOWLEDGE IN DIFFUSION MODELS

**Koichi Namekata**[1,2]**, Amirmojtaba Sabour**[1,2]**, Sanja Fidler**[1,2,3]**, Seung Wook Kim**[1,2,3]
[1]University of Toronto, [2]Vector Institute, [3]NVIDIA
koichi.namekata@mail.utoronto.ca, {amsabour, fidler, seung}@cs.toronto.edu

## ABSTRACT

Diffusion models have recently received increasing research attention for their re-markable transfer abilities in semantic segmentation tasks. However, generating fine-grained segmentation masks with diffusion models often requires additional training on annotated datasets, leaving it unclear to what extent pre-trained dif-fusion models alone understand the semantic relations of their generated images. To address this question, we leverage the semantic knowledge extracted from Sta-ble Diffusion (SD) and aim to develop an image segmentor capable of generating fine-grained segmentation maps without any additional training. The primary dif-ficulty stems from the fact that semantically meaningful feature maps typically exist only in the spatially lower-dimensional layers, which poses a challenge in directly extracting pixel-level semantic relations from these feature maps. To overcome this issue, our framework identifies semantic correspondences between image pixels and spatial locations of low-dimensional feature maps by exploiting SD's generation process and utilizes them for constructing image-resolution seg-mentation maps. In extensive experiments, the produced segmentation maps are demonstrated to be well delineated and capture detailed parts of the images, in-dicating the existence of highly accurate pixel-level semantic knowledge in diffu-sion models. *Project page:* https://kmcode1.github.io/Projects/ EmerDiff/

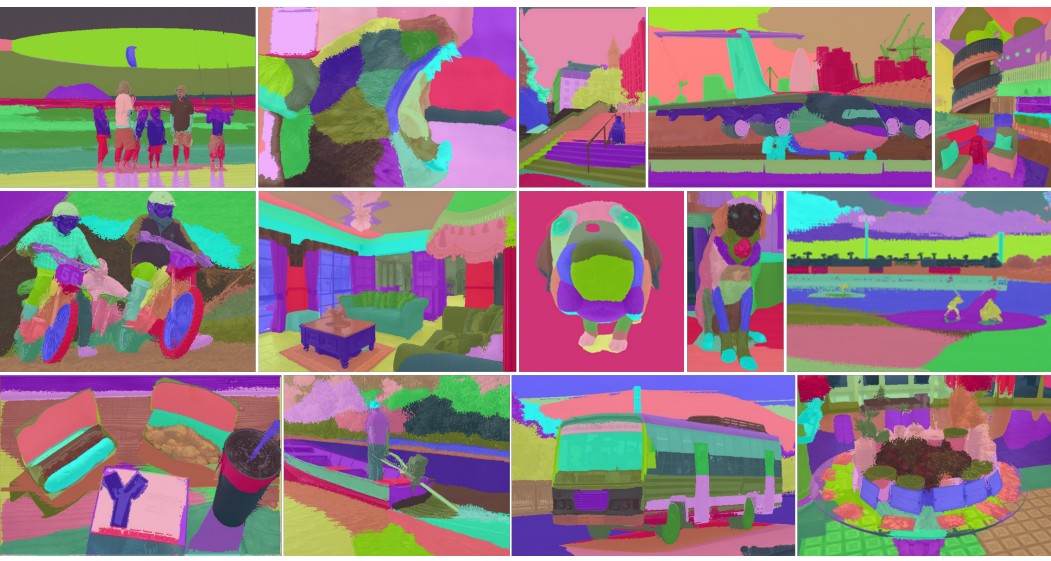

Figure 1: EmerDiff is an unsupervised image segmentor solely built on the semantic knowledge extracted from a pre-trained diffusion model. The obtained fine-detailed segmentation maps suggest the presence of highly accurate pixel-level semantic knowledge in diffusion models.

# 1 INTRODUCTION

In recent years, diffusion models (Ho et al., 2020; Dhariwal & Nichol, 2021) have emerged as the state-of-the-art generative models for synthesizing high-quality images. Notably, the internal representations of pre-trained diffusion models have been found semantically enriched, demonstrating impressive transfer abilities in semantic segmentation tasks (Baranchuk et al., 2022; Karazija et al., 2023; Li et al., 2023a; Xu et al., 2023a). However, the previous success in producing fine-grained segmentation maps often relies on incorporation of additional knowledge such as mask annotations (Baranchuk et al., 2022; Xu et al., 2023a) and hand-crafted priors (Karazija et al., 2023; Ma et al., 2023; Wu et al., 2023), leaving it unclear to what extent the pre-trained diffusion models alone understand the semantic relations of their generated images. To address this question, we present an unsupervised image segmentor that can generate fine-grained segmentation maps by solely leveraging the semantic knowledge extracted from a pre-trained diffusion model. Specifically, our work is built on Stable Diffusion (Rombach et al., 2022), a large-scale text conditioned diffusion model capable of generating diverse high-resolution images.

The common unsupervised approach to capture semantic relations from a diffusion model is to apply k-means on its semantically meaningful feature maps. This technique has been demonstrated to produce cluster maps that are semantically aligned (Baranchuk et al., 2022; Patashnik et al., 2023; Xu et al., 2023a). While this method intuitively visualizes the semantic awareness of diffusion models, the obtained semantic relations are often coarse because of the fact that the semantically meaningful feature maps typically reside only in the spatially low-dimensional layers (Collins et al., 2020; Baranchuk et al., 2022; Luo et al., 2023). Nevertheless, diffusion models possess the ability to construct high-resolution images based on their semantic knowledge embedded in low-resolution feature maps. This motivates us to analyze how the semantically meaningful low-resolution feature maps influence the output images through the generation process, which we hypothesize to be a crucial step for extracting pixel-level semantic knowledge from the diffusion models.

To this end, we investigate the influence of local change in the values of the low-resolution feature maps on the pixel values of the generated images. Our key discovery is that when we perturb the values of a sub-region of low-resolution feature maps, the generated images are altered in a way that only the pixels semantically related to that sub-region are notably changed. Consequently, we can automatically identify the semantic correspondences between image pixels and a sub-region of low-dimensional feature maps by simply measuring the change in the pixel values.

Building on this insight, our proposed image segmentor can generate fine-grained segmentation maps without the need for any additional knowledge. First, we generate *low-resolution segmentation maps* (e.g., $16 \times 16$) by applying k-means on low-dimensional feature maps. Then, we build *image-resolution segmentation maps* (e.g., $512 \times 512$) in a top-down manner by mapping each image pixel to the most semantically corresponding low-resolution mask. These semantic correspondences are extracted from the diffusion models leveraging the aforementioned finding.

The effectiveness of our framework is extensively evaluated on multiple scene-centric datasets such as COCO-Stuff (Caesar et al., 2018), PASCAL-Context (Mottaghi et al., 2014), ADE20K (Zhou et al., 2019) and Cityscapes (Cordts et al., 2016) both qualitatively and quantitatively. Although the underlying diffusion model is never trained on annotated datasets, our framework produces segmentation maps that align surprisingly well with the detailed parts of the images, indicating the existence of highly accurate pixel-level semantic knowledge in the diffusion models.

# 2 RELATED WORKS

**Generative models for semantic segmentation.** The use of generative models for semantic segmentation dates back to the era of GANs, where Collins et al. (2020) discovered applying k-means on the StyleGAN's (Karras et al., 2019) intermediate feature maps yields clusters aligning well with semantic objects. Following that, the prior works (Li et al., 2021; Tritrong et al., 2021; Xu & Zheng, 2021; Zhang et al., 2021; Li et al., 2022b) exploited such semantically meaningful feature maps to learn semantic segmentation with minimal supervision. Notably, diffusion models exhibit similar properties, where clusters of their intermediate feature maps consistently group semantic objects (Baranchuk et al., 2022; Xu et al., 2023a). These feature maps are utilized in various downstream tasks, including keypoint matching (Hedlin et al., 2023; Luo et al., 2023; Tang et al.,

2023a; Zhang et al., 2023) and semantic segmentation (Baranchuk et al., 2022; Li et al., 2023a; Xu et al., 2023a), outperforming GAN-based counterparts. Additionally, cross-attention layers of text-conditioned diffusion models (Balaji et al., 2022; Rombach et al., 2022; Saharia et al., 2022) are used for determining object layouts (Hertz et al., 2022; Patashnik et al., 2023; Tang et al., 2023b), where delimited layouts serve as (pseudo-) segmentation masks (Karazija et al., 2023; Ma et al., 2023; Wu et al., 2023). However, such semantically meaningful feature maps usually exist in low-dimensional layers, whose spatial resolution is significantly lower than the image resolution. To obtain upsampled and refined segmentation maps, prior literature has incorporated post-processing tools such as boundary refinement techniques (Krähenbühl & Koltun, 2011; Barron & Poole, 2016; Araslanov & Roth, 2020; Wang et al., 2023) which, however, relies on hand-crafted priors. In contrast, our framework successfully produces fine-grained segmentation masks (e.g., $512 \times 512$) from low-resolution feature maps (e.g., $16 \times 16$) without the need for any additional knowledge.

**Unsupervised semantic segmentation.** Unsupervised semantic segmentation is the task of grouping pixels of unlabeled images into semantically meaningful concepts without seeing annotated datasets. Previous studies have focused on either segmenting only salient objects (Van Gansbeke et al., 2021; Melas-Kyriazi et al., 2022a; Shin et al., 2022; Wang et al., 2022; Siméoni et al., 2023; Zadaianchuk et al., 2023) or segmenting entire scenes (Ji et al., 2019; Cho et al., 2021; Hamilton et al., 2022; Seitzer et al., 2022; Wen et al., 2022; Yin et al., 2022; Li et al., 2023b), where our study is of the latter category. Typically, those frameworks are composed of two parts: 1. train an image encoder that produces pixel embeddings through self-supervised learning. 2. learn concept embeddings that are used to group pixels into a pre-defined number of semantic concepts. During inference, the pixels are classified into one of these concepts by matching their pixel embeddings with the closest concept embeddings. The current state-of-the-art approaches for scene segmentation (Hamilton et al., 2022; Seitzer et al., 2022; Wen et al., 2022) are built on the pre-trained self-supervised ViTs like DINO (Caron et al., 2021). There also exist several studies that utilize GANs' latent spaces for segmenting foreground objects (Voynov et al., 2020; Abdal et al., 2021; Melas-Kyriazi et al., 2022b; Feng et al., 2023; Oldfield et al., 2023); however, these approaches only work in narrow visual domains and are not applicable to scene-centric images.

**Open-vocabulary semantic segmentation.** Open-vocabulary semantic segmentation aims to segment images according to arbitrary user-defined vocabularies during inference. Models are typically trained with only text-image pairs (Xu et al., 2022a; Zhou et al., 2022; Cha et al., 2023; Mukhoti et al., 2023; Ranasinghe et al., 2023; Xu et al., 2023b), or combination of unlabeled/labeled annotations and text-image supervision (Ghiasi et al., 2022; Li et al., 2022a; Xu et al., 2022b; Liang et al., 2023; Xu et al., 2023a;c). The majority of these models are built on image encoders of pre-trained vision language models like CLIP (Radford et al., 2021), but learn to produce feature representations that exhibit better pixel-level alignment with text embeddings. However, the existing annotation-free models, which are trained without annotated datasets, tend to produce noisy segmentation masks. To overcome this issue, we integrate our framework into these models. Concretely, for each segmentation mask produced from our framework, we compute its mask embedding through their image encoders and classify them by the text embeddings. By combining our framework's segmentation abilities with their classification abilities, we achieve significantly better mIoU.

## 3 METHODS

As illustrated in Figure 2, our goal is to generate fine-grained segmentation maps by solely leveraging the semantic knowledge extracted from pre-trained diffusion models. To achieve this, we begin by generating *low-resolution segmentation maps* by applying k-means on the semantically meaningful low-dimensional feature maps (Section 3.2). Next, we construct image-resolution segmentation maps by mapping each image pixel to the most semantically corresponding low-resolution mask. To find the semantic correspondences between the image pixels and the masks, we exploit the diffusion model's mechanism of generating high-resolution images from their low-resolution feature maps (Section 3.3). In the following sections, we first provide an overview of the properties of diffusion models (Section 3.1) and then delve into further details of our approach.

### 3.1 PRELIMINARIES

Diffusion models are trained to generate images by taking successive denoising steps from pure Gaussian noise, where each denoising step is commonly performed with U-Net backbones (Ron-

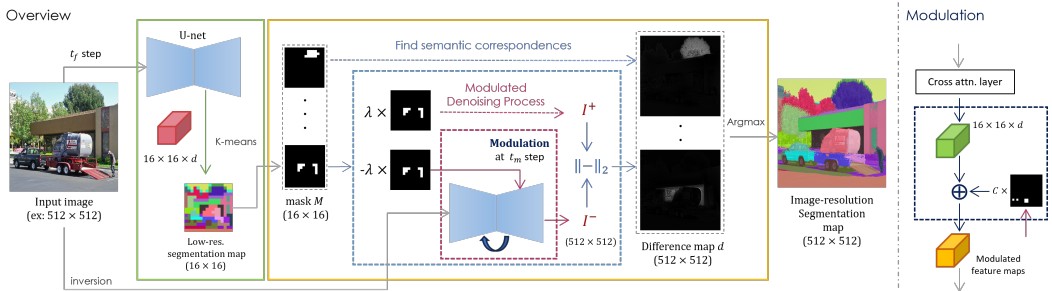

Figure 2: **Overview of our framework.** *green*: we first construct low-resolution segmentation maps by applying k-means on semantically meaningful low-dimensional feature maps. *orange*: Next, we generate image-resolution segmentation maps by mapping each pixel to the most semantically corresponding low-resolution mask, where semantic correspondences are identified by the modulated denoising process.

neberger et al., 2015) containing downward and upward paths. Specifically, our framework is built on Stable Diffusion (SD) (Rombach et al., 2022), where the denoising steps are performed in a spatially lower-dimensional latent space, and final generated images are obtained by decoding denoised latents with another neural network. The SD's U-Net architecture takes both noisy latents and text captions (an empty string in our experiments) as inputs and processes them through a stack of *modular blocks*, where each block consists of a residual block (He et al., 2016), a self-attention layer and a cross-attention layer (Vaswani et al., 2017). These blocks belong to one of the four spatial resolution levels (namely, $8 \times 8$, $16 \times 16$, $32 \times 32$, and $64 \times 64$), where the components of our interest are modular blocks at resolution $16 \times 16$ on the upward path, which especially contain semantically enriched representations (Luo et al., 2023) and are utilized in the prior literature (Karazija et al., 2023; Patashnik et al., 2023). Specifically, there are three consecutive modular blocks at resolution $16 \times 16$ on the upward path, where Voynov et al. (2023) have observed that the first cross-attention layer is primarily responsible for manipulating contents, while the last cross-attention layer exerts more control over appearance. In the subsequent sections, we make use of these semantically meaningful layers to produce fine-grained segmentation maps.

## 3.2 Constructing low-resolution segmentation maps

To handle real images, we first invert real images into a particular number of denoising steps (each specified by timestep $t = 1 \cdots T$ where larger timesteps correspond to noisier images) through DDPM-based inversion (Huberman-Spiegelglas et al., 2023), which guarantees perfect reconstruction with the scheduled noise. Next, we extract query vectors from the first cross-attention layer of upward $16 \times 16$ modular blocks at timestep $t_f$, which will be referred to as our low-dimensional feature maps. Intuitively, query vectors are trained to directly interact with text tokens; hence, their representations should be semantically aware. Finally, we apply k-means on the extracted feature maps, obtaining $K$ clusters serving as *low-resolution segmentation masks*.

## 3.3 Building image-resolution segmentation maps

Thus far, we have constructed low-resolution segmentation maps (e.g., $16 \times 16$), which are 32 times lower in resolution than the original image (e.g., $512 \times 512$). Our next goal is to build image-resolution segmentation maps from low-resolution segmentation maps by identifying the semantic correspondences between the image pixels and the low-resolution masks. To this end, we begin by observing how the low-dimensional layers in SD influence the pixels of their generated images. Specifically, we *modulate* the values of a sub-region of feature maps at $16 \times 16$ cross-attention layers and observe how this local change effects the pixel values of the generated images.

In the official SD's implementation, a cross-attention layer projects inputs into query, key, and value vectors (denote $Q, K, V$) and computes $f\left(\sigma\left(\frac{QK^T}{\sqrt{d}}\right) \cdot V\right) \in \mathbb{R}^{hw \times d}$, where $d$ is the dimension of query vectors, $hw$ is the spatial dimension of modular blocks, $\sigma$ is a softmax function, and $f$ is a fully-connected layer. To modulate the cross-attention layer, we replace this computation by

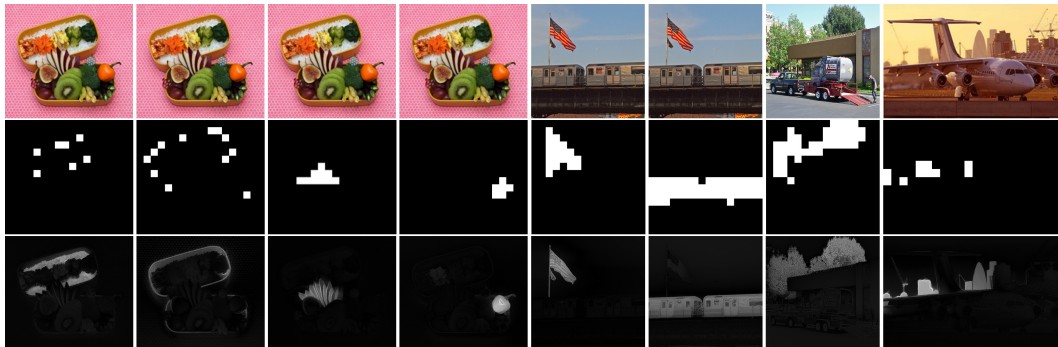

Figure 3: **Visualization of modulated denoising process.** First row: original image. Second row: low-resolution modulation mask $M \in \{0,1\}^{h \times w}$. Third row: obtained difference map $d \in \mathbb{R}^{H \times W}$, where $H/h = W/w = 32$

$f\left(\sigma\left(\frac{QK^T}{\sqrt{d}}\right) \cdot V\right) + cM \in \mathbb{R}^{hw \times d}$ where $c \in \mathbb{R}$ is a scalar value and $M \in \{0,1\}^{hw \times 1}$ is a binary mask specifying the spatial locations to modulate. Verbally, we uniformly add a constant offset $c$ to a sub-region (specified by $M$) of feature maps. During the *modulated denoising process*, we apply this modulation to a specific cross-attention layer at timestep $t_m$.

To observe changes in pixel values, we first run the above modulated denoising process with two offsets $c = -\lambda, +\lambda$ separately, obtaining two altered images $I^-, I^+ \in \mathbb{R}^{H \times W \times 3}$. We then compute the difference maps $d$ by taking Euclidean distance over RGB dimension $d = ||I^- - I^+||_2 \in \mathbb{R}^{H \times W}$. As shown in Figure 3, the pixels semantically related to the modulated sub-region changed prominently, while the other pixels remained roughly the same. Therefore, the obtained difference map can be interpreted as the *strength* of the semantic correspondences between the image pixels and the sub-region of the low-resolution feature maps.

Based on this observation, we can compute the strength of semantic correspondences between every pair of image pixels and the low-resolution segmentation masks. In detail, for each low-resolution mask $M^i \in \{0,1\}^{hw \times 1}$, we produce a difference map $d^i \in \mathbb{R}^{H \times W}$, where $d^i_{x,y} \in \mathbb{R}$ represents the strength of semantic correspondence between pixel $(x,y)$ and mask $i$. To build image-resolution segmentation maps, we label each pixel $(x,y)$ with the low-resolution mask $k$ having the strongest semantic correspondence (i.e., $k = \mathrm{argmax}_i d^i_{x,y}$).

For further improvements, we fix the attention maps $QK^T$ (i.e., inject original attention maps) of all the self/cross attention layers during the modulated denoising process. Since the attention maps represent pixel affinities and strongly influence object layouts, attention injection is a commonly used technique in image editing to preserve the structure of images (Tumanyan et al., 2023). Finally, after computing the difference maps, we apply Gaussian filtering to suppress pixelated artifacts.

## 4 EXPERIMENTS

In this section, we evaluate our produced segmentation masks both qualitatively and quantitatively. We also conduct hyperparameter analysis in Appendix D.

### 4.1 IMPLEMENTATION DETAILS

Throughout the experiments, we use the official Stable Diffusion v1.4 checkpoint with DDPM sampling scheme of 50 steps (for clarity purposes, we denote timesteps out of $T = 1000$). To generate low-resolution segmentation maps, we extract feature maps at timestep $t_f = 1$ (minimum noise). We apply modulation to the third cross-attention layer of $16 \times 16$ upward blocks at timestep $t_m = 281$ and $\lambda = 10$. As discussed earlier, this layer is responsible for controlling appearance. The effects of varying hyperparameters are discussed in Appendix D.

**Runtime analysis.** The most computationally expensive part of our method is the modulated denoising process, which runs independently for each mask. However, we only need to execute the modulated denoising process from the timestep to apply modulation (i.e. modulation timestep $t_m$),

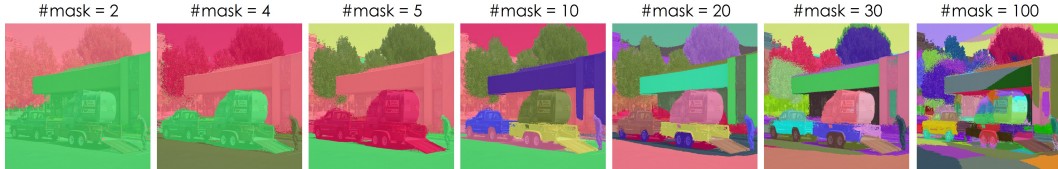

Figure 4: Qualitative comparison with naively upsampled low-resolution segmentation maps.

Figure 5: **Varying the number of segmentation masks.** Our framework consistently groups objects in a semantically meaningful manner.

Table 1: **Results of unsupervised semantic segmentation under traditional evaluation strategy.** Evaluated on full COCO-Stuff-27 (Caesar et al., 2018). ACSeg is taken from the original paper. IIC, PiCIE, and TransFGU from Yin et al. (2022). DINO from Koenig et al. (2023). Other results from Seitzer et al. (2022). Some works are evaluated on curated datasets (Ji et al., 2019), which generally gives higher mIoU than being evaluated on the full datasets (Yin et al., 2022).

| | mIoU (↑) | | mIoU (↑) |
|---|---|---|---|
| IIC (Ji et al., 2019) | 2.4 | TransFGU (Yin et al., 2022) | 16.2 |
| PiCIE (Cho et al., 2021) | 11.9 | SlotCon (Wen et al., 2022) | 18.3 |
| DINO ViT-B/8 (Caron et al., 2021) | 13.0 | STEGO (Hamilton et al., 2022) | **26.8** |
| SegDiscover (Huang et al., 2022) | 14.3 | DINOSAUR (Seitzer et al., 2022) | 24.0 |
| ACSeg (Li et al., 2023b) | 16.4 | **Ours** | 26.6 |

requiring 15 denoising steps in our hyperparameter settings. Furthermore, our method does not involve backpropagation, and the entire denoising process is operated in the latent space (except the last decoding part). Therefore, parallelization can be performed effectively.

## 4.2 QUALITATIVE ANALYSIS

Figure 1 showcases the examples of our produced segmentation maps (More results in Appendix E). Our segmentation maps are well delineated and successfully distinguish moderately small objects (e.g., person, building) in the scene. To further demonstrate the effectiveness of our pipeline, we compare our segmentation maps with the naively up-sampled (via bilinear interpolation) low-resolution segmentation maps. As visualized in Figure 4, the naively upsampled segmentation maps are coarse and hard to interpret. In contrast, our segmentation maps are much clearer despite sharing the same low-resolution maps. Finally, we vary the number of masks generated per image. As illustrated in Figure 5, we obtain segmentation maps that are semantically interpretable even when the number of masks is as small as 2.

## 4.3 QUANTITATIVE ANALYSIS

To quantitatively evaluate our segmentation masks, we apply our framework to two downstream tasks: unsupervised semantic segmentation and annotation-free open vocabulary segmentation. In this evaluation, our framework generates 30 segmentation masks per image.

**Unsupervised semantic segmentation.** We evaluate our framework on standard segmentation datasets. The traditional evaluation protocol requires pixels to be classified into the same number of semantic concepts as the dataset classes so that the concepts can be subsequently matched with the dataset classes through Hungarian matching. As in prior work (Hamilton et al., 2022), we extend our framework by generating pixel/concept embeddings, where each pixel will be classified into the concept with the closest embeddings. To generate pixel embeddings, we first create a mask

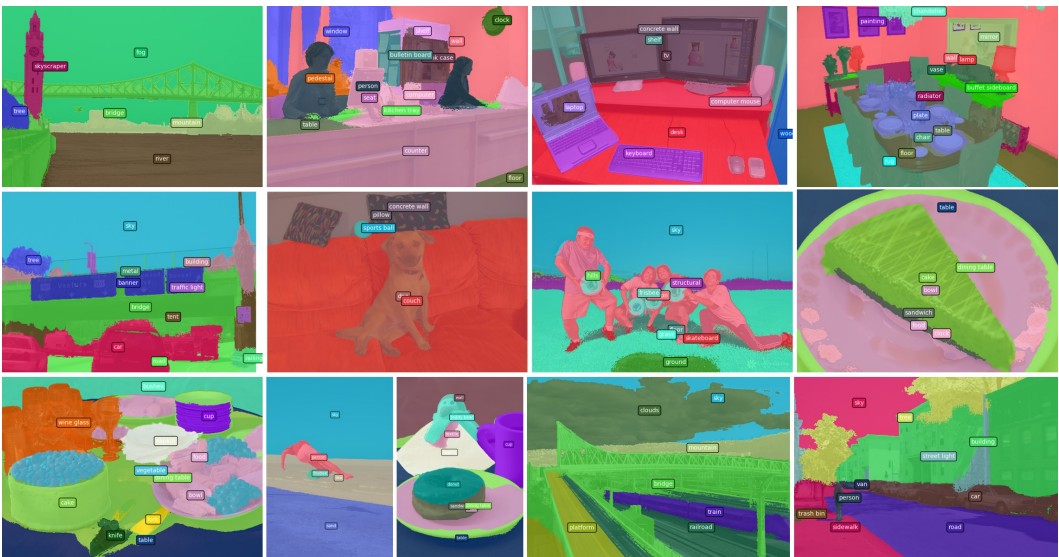

Figure 6: **Visualizations of unsupervised semantic segmentation under our modified evaluation strategy.** More examples in Figure 11 in Appendix.

Table 2: **Results of unsupervised semantic segmentation under our modified evaluation strategy.** Evaluated on ADE20K (AD150) (Zhou et al., 2019), PASCAL-Context (PC59, PC459) (Mottaghi et al., 2014), COCO-Stuff (CS171, CS27) (Caesar et al., 2018), and Cityscapes (City19) (Cordts et al., 2016). MDC (Cho et al., 2021), PiCIE (Cho et al., 2021), DINO, and STEGO are trained solely on images, while CLIP (Radford et al., 2021), TCL (Cha et al., 2023), and CLIPpy (Ranasinghe et al., 2023) are trained on text-image pairs. For CLIP, we follow Zhou et al. (2022) to modify the image encoder to output pixel-wise embeddings. For SD, we naively up-sample low-resolution segmentation maps (via bilinear interpolation, see Figure 4) and produce pixel embeddings following the same procedure as ours.

|  |  | mIoU(↑) | | | | | |
| --- | --- | --- | --- | --- | --- | --- | --- |
|  | Backbone | AD150 | PC59 | PC459 | CS171 | CS27 | City19 |
| MDC | - | - | - | - | - | 8.6 | 14.0 |
| PiCIE | - | - | - | - | - | 11.7 | 15.1 |
| DINO | DINO ViT-B/8 | 19.1 | 30.0 | 10.1 | 19.0 | 32.2 | 34.6 |
| STEGO | DINO ViT-B/8 | - | - | - | 13.8 | 36.6 | 34.6 |
| CLIP | CLIP ViT-B/16 | 22.0 | 36.4 | 14.0 | 23.9 | 34.1 | 33.7 |
| TCL | CLIP ViT-B/16 | 20.8 | 35.5 | 11.3 | 23.5 | 31.9 | 32.6 |
| CLIPpy | T5 + DINO ViT-B/16 | 24.0 | 38.0 | 15.6 | 25.9 | 34.1 | 28.3 |
| SD | SD v1.4 | 29.1 | 41.5 | 20.6 | 27.6 | 42.1 | 32.3 |
| **Ours** | SD v1.4 | **33.1** | **45.7** | **25.1** | **30.5** | **45.8** | **37.1** |

embedding of each segmentation mask by utilizing SD's low-dimensional feature maps (details are in appendix B). Each pixel then adopts its corresponding mask embedding as its own pixel embedding. For concept embeddings, we run k-means on the pixel embeddings across the entire dataset and extract the desired number of cluster centroids.

However, as demonstrated in Table 1, we observe that our model performs on par with the recent DINO-based baselines: STEGO (Hamilton et al., 2022) and DINOSAUR (Seitzer et al., 2022). This is attributed to the limitation of the traditional evaluation protocol that mIoU is sensitive to how well the concept embeddings align with the pre-defined dataset classes, which is problematic when there are various ways to semantically group objects into a fixed number of groups (See Figure 9 in Appendix for concrete examples).

To demonstrate the strength of our framework, we relax the restrictions of the traditional evaluation protocol by allowing access to the annotations while building concept embeddings. Specifically, we take the average of pixel embeddings belonging to the same ground truth labels and set the obtained

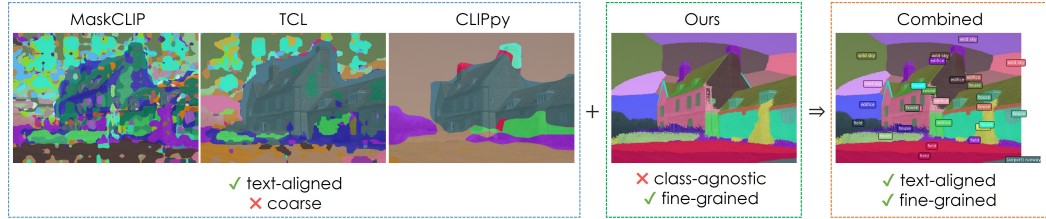

Figure 7: **Annotation-free open-vocabulary semantic segmentation.** We combine our fine-grained class-agnostic segmentation masks (green) with the baseline's coarse text-aligned pixel embeddings (blue) to produce text-aligned fine-grained segmentation maps (orange).

Table 3: **Comparison between baselines and baselines + ours in annotation-free open vocabulary semantic segmentation.** Evaluated on ADE20K(AD150) (Zhou et al., 2019), PASCAL-Context(PC59, PC459) (Mottaghi et al., 2014), COCO-Stuff(CS171) (Caesar et al., 2018), and Cityscapes(City19) (Cordts et al., 2016). For a fair comparison, we re-evaluate TCL, MaskCLIP, and CLIPpy with the same prompt engineering. The results of other works are also put for reference, where OVSegmentor is taken from the original paper, and GroupViT from Cha et al. (2023)

| | mIoU (↑) | | | | |
| | AD150 | PC59 | PC459 | CS171 | City19 |
|---|---|---|---|---|---|
| OVSegmentor (Xu et al., 2023b) | 5.6 | 20.4 | - | - | - |
| GroupViT (Xu et al., 2022a) | 9.2 | 23.4 | - | 15.3 | 11.1 |
| MaskCLIP (Zhou et al., 2022) | 11.5 | 25.4 | 4.25 | 15.3 | 20.9 |
| MaskCLIP + **Ours** | **15.9** | **33.2** | **6.54** | **20.7** | **26.5** |
| TCL (Cha et al., 2023) | 14.6 | 30.9 | 5.22 | 18.7 | 20.9 |
| TCL + **Ours** | **17.4** | **35.4** | **6.49** | **21.8** | **23.4** |
| CLIPpy (Ranasinghe et al., 2023) | 12.6 | 28.0 | 4.64 | 16.6 | **9.76** |
| CLIPpy + **Ours** | **12.9** | **29.0** | **4.88** | **17.2** | 9.24 |

mean embeddings as concept embeddings. As illustrated in Figure 6, this modification ensures that concept embeddings represent each dataset class, and mIoU difference primarily comes from the accuracy and disentanglement capabilities of pixel embeddings. Since our modified evaluation protocol only requires a model outputting pixel-wise embeddings, it can be applied to other baselines, including STEGO (Hamilton et al., 2022) as well as self-supervised image encoders.

As presented in Table 2, our modified evaluation protocol reveals a clear difference between the previous methods and our framework. We observe that the DINO-based models (DINO and STEGO) generally perform poorly when the number of classes in a dataset is large, while the diffusion-based models (SD and ours, where SD is the naively up-sampled low-resolution masks as in Figure 4) handle them relatively well. This might be because DINO is originally trained on curated object-centric datasets like ImageNet (Russakovsky et al., 2015), whereas SD is trained on diverse images, making itself more robust to various domains. Furthermore, we observe that STEGO performs on par with DINO, which is consistent with the follow-up results (Koenig et al., 2023) that STEGO has similar or even inferior linear probing performance to DINO. Nevertheless, our method consistently outperforms all the baselines. Additionally, we conduct comparisons with text-aligned image encoders (CLIP, TCL, CLIPpy) and still observe the performance gaps, attesting our performance gains are not solely attributed to the availability of text-image pairs during the pre-training. Lastly, we see a clear improvement between our framework and SD, where they share the same mask embeddings and only differ in mask shapes. This validates the effectiveness of our pipeline.

**Annotation-free open-vocabulary semantic segmentation.** In this study, we combine our framework with the existing annotation-free open-vocabulary segmentation models, where both approaches are trained on image-text pairs only without access to annotated datasets. As visualized in Figure 7, the existing baselines produce text-aligned but coarse pixel embeddings, while our framework produces well delineated but class-agnostic masks. This motivates us to combine our segmentation masks with their pixel embeddings, aiming to produce class-aware fine-grained segmentation masks.

Figure 8: **Failure cases.** Segmentation masks occasionally failed to distinguish extremely small objects (e.g., small desks, animal legs, human's face parts).

In detail, we integrate our framework into three publicly available annotation-free baselines containing image and text encoders: MaskCLIP (Zhou et al., 2022), TCL (Cha et al., 2023), and CLIPpy (Ranasinghe et al., 2023). MaskCLIP (without heuristic refinement) is the most standard baseline, where it modifies the pre-trained CLIP image encoder (Radford et al., 2021) to output pixel-wise embeddings without additional training. TCL and CLIPpy are structured similarly to MaskCLIP, but trained to produce better pixel-level representations. To combine our framework with these baselines, we first generate mask embeddings of each mask by computing the average of pixel embeddings produced from the baseline's image encoder. We then classify each mask by finding the closest text embedding to its mask embedding. Following convention Radford et al. (2021), prompt templates (e.g., *"A photo of a {}"*) are used to produce text embeddings. We provide the list of templates in Appendix C.

As presented in Table 3, we mostly observe the performance gain after being combined with our framework, assuring the quality of our segmentation masks. Notably, the performance gain of MaskCLIP is substantial, exhibiting competitive performance with the recent baselines (before being combined with our method). On the other hand, the performance gain of CLIPpy is marginal. We attribute this to their over-smoothed pixel embeddings (See Figure 7).

## 5 LIMITATION AND CONCLUSION

In this paper, we developed an unsupervised image segmentor that can generate fine-grained segmentation maps by solely leveraging the semantic knowledge extracted from a pre-trained diffusion model. The extensive experiments validated its effectiveness, suggesting the presence of highly accurate pixel-level semantic knowledge in diffusion models.

As a limitation, our framework occasionally struggles to distinguish extremely small objects (e.g., animal legs, human faces) as illustrated in Figure 8, which might be because the detailed parts are compressed together in low-dimensional layers, and our framework fails to separate them when generating low-resolution segmentation maps. Additionally, the underlying feature representations may contain not only object meanings but also other attributes such as spatial location and colors, which lead some objects such as sky and ground to be over-segmented. For practical use, treating our produced masks as pseudo-masks and integrating them into weakly-supervised frameworks could be a promising direction.

Lastly, our study is built on Stable Diffusion, since it is the most commonly studied diffusion model that can generate diverse images. However, our fundamental idea of modulating semantically meaningful feature maps can be potentially applied to various generative models, which we leave for future work. We hope that our findings help further understand the inner workings of diffusion models and also encourage the research direction that utilizes generative models for discriminative tasks.

## ACKNOWLEDGEMENTS

The authors acknowledge support by NSERC and the Vector Institute. SF acknowledges the Canada CIFAR AI Chair award.

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

## A    ADDITIONAL IMPLEMENTATION DETAILS

In order to process real images with SD, each dimension of images is required to be a multiple of $64$. Furthermore, the size of images should be adjusted so that it won't be too large or too small to process with SD. To meet these requirements, we resize the images to keep pixel count roughly $512^2$ while keeping the original height-to-width ratio. We then round up each dimension to the nearest integer divisible by $64$. When computing mIoU, the obtained segmentation maps are resized back to the original size via nearest neighboring.

## B    ADDITIONAL DETAILS OF UNSUPERVISED SEMANTIC SEGMENTATION

**Implementation details.** The traditional evaluation protocol (Ji et al., 2019) requires that the model classifies pixels into the same number of semantic concepts as the dataset classes so that it can subsequently match the semantic concepts with the pre-defined dataset classes through Hungarian matching (Here we maximize mIoU). To fulfill this requirement, previous works learn both pixel embeddings and concept embeddings so that it can classify each pixel into the concept having the closest concept embedding from the pixel embedding. In particular, the current state-of-the-art model, STEGO (Hamilton et al., 2022), first trains an image encoder that produces pixel embeddings

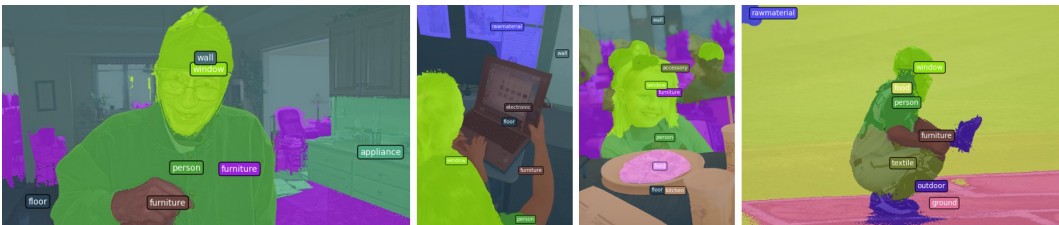

Figure 9: **Drawbacks of the traditional evaluation protocol of unsupervised semantic segmentation.** Evaluated on COCO-Stuff-27. Our framework partitions a person into head, arm, and body but there is only one pre-defined class for *person* in the dataset, forcing other parts to be mismatched with irrelevant classes.

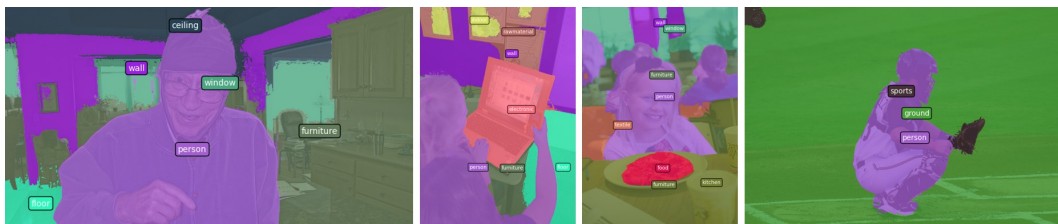

Figure 10: **Unsupervised semantic segmentation under our modified unsupervised semantic segmentation evaluation protocol.** Evaluated on COCO-Stuff-27. In contrast to Figure 9, our framework partitions objects following the pre-defined classes in the dataset (e.g., one group for a person), evaluating the quality of our segmentaion masks more appropriately.

and then clusters pixel embeddings over the entire dataset, obtaining cluster centroids that serve as the concept embeddings.

To adapt our framework to this task, we follow STEGO's procedure and extend our framework by generating pixel/concept embeddings. To create pixel embeddings, we generate a mask embedding for each produced mask using SD's feature maps. Specifically, we generate a mask embedding $e \in \mathbb{R}^c$ of a mask $M \in \{0, 1\}^{H \times W}$ by taking the average of SD's feature maps $F \in \mathbb{R}^{c \times h \times w}$ inside the masked region $m \in \{0, 1\}^{h \times w}$ where $F$ is the query vectors extracted from the first cross-attention layer of upward $16 \times 16$ at timestep $t = 200$, and $m$ is the $M$'s low-resolution mask. Note that these designs are largely borrowed from the previous work (Karazija et al., 2023) that produces mask embeddings using SD's feature maps for classification purposes. Once the mask embeddings are generated, each pixel embedding is defined as the embedding of the mask it belongs to. To generate concept embeddings under the traditional evaluation protocol, we run k-means on the pixel embeddings across the entire dataset and extract their cluster centroids.

**Issues of traditional evaluation protocol.** To attain high mIoU in this evaluation protocol, it unwillingly becomes crucial to generate concept embeddings that align well with the pre-defined dataset classes. Since we are not allowed to access ground truth labels for building concept embeddings, this requirement becomes unreasonably strict and makes evaluation unreliable, especially when there are multiple valid ways to semantically partition objects. For instance, COCO-Stuff 27 (Caesar et al., 2018) is the popular dataset used in unsupervised semantic segmentation, where it only consists of one class referring to *person*; however, as visualized in Figure 9, our framework produces at least three semantic concepts for a person: head, body, and arms. Since only one concept can be matched with the person class, other parts are forced to be mismatched with irrelevant labels, resulting in a decrease in mIoU.

**Modifying traditional evaluation protocol.** Since our focus is not on estimating dataset classes but on properly evaluating the quality of our produced segmentation masks, we pre-calculate concept embeddings such that each embedding corresponds to each dataset class. To achieve this, we relax the restriction of the traditional evaluation protocol by allowing access to the ground truth annotations while constructing concept embeddings. Specifically, for each dataset class, we aggregate all

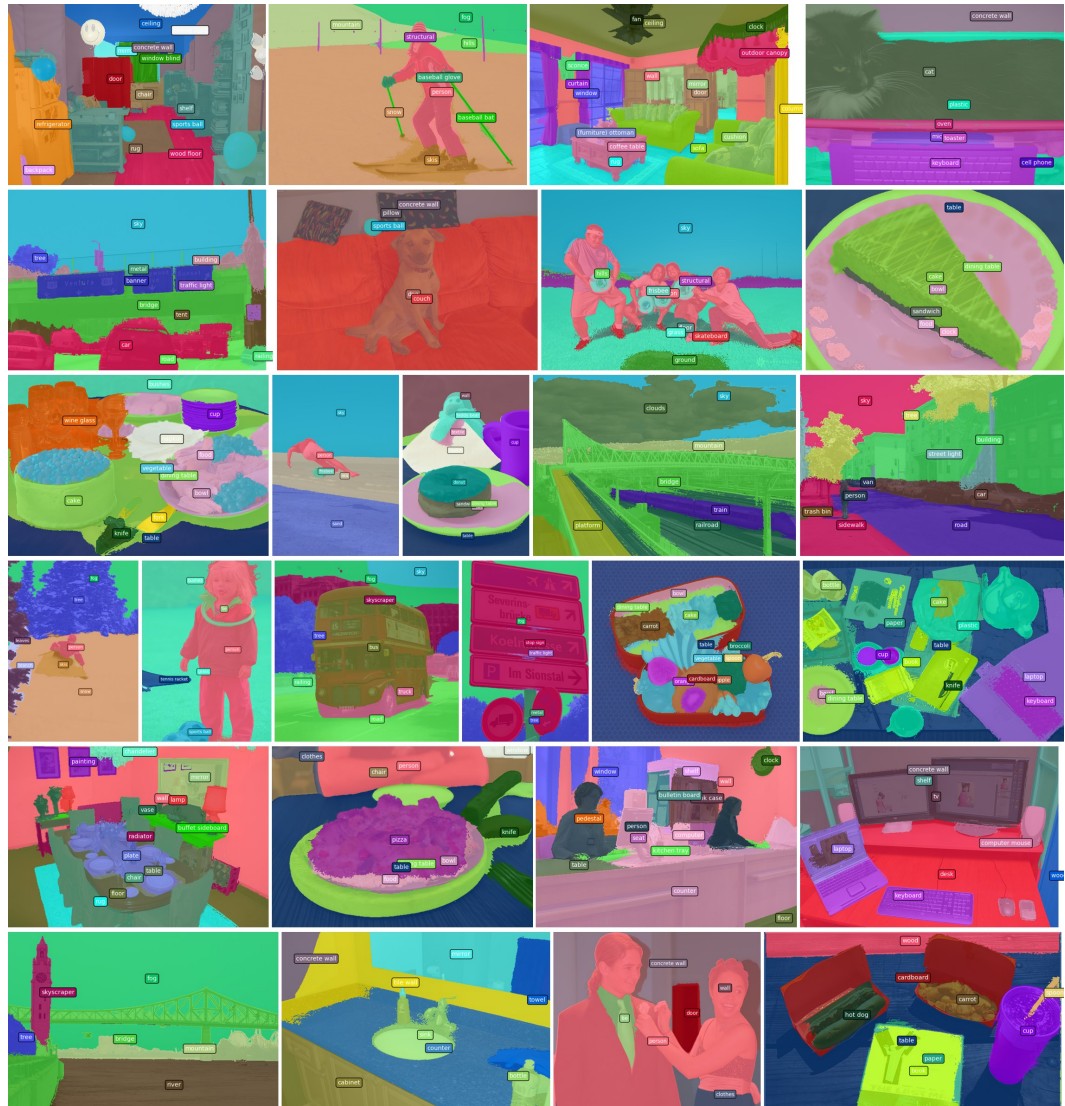

Figure 11: Examples of unsupervised semantic segmentation on COCO171 and ADE150K, evaluated under our modified evaluation protocol.

pixel embeddings belonging to the class from the entire dataset and compute their average. The obtained mean embedding serves as the concept embedding of this class. As visualized in Figure 10, this modified evaluation protocol enables our framework to partition the objects according to the pre-defined class categories in the dataset, and the performance difference primarily comes from the quality of underlying pixel embeddings rather than the concept embeddings. Additional qualitative results can be found in Figure 11.

## C ADDITIONAL DETAILS OF OPEN-VOCABULARY SEMANTIC SEGMENTATION

**Implementation details.** As discussed in Cha et al. (2023), there exists no unified evaluation protocol for open-vocabulary semantic segmentation, and different studies often employ varying prompt engineering strategies. For a fair comparison, we re-evaluated our baselines with the same class labels and prompt templates. Specifically, following Mukhoti et al. (2023), we use 7 prompt templates: *"itap of a {}.", "a bad photo of a {}.", "a origami {}.", "a photo of the large {}.", "a {} in a video game.","art of the {}.", "a photo of the small {}.",* where {} is replaced with the class

Table 4: Effects of varying the number of masks on open-vocabulary segmentation and unsupervised semantic segmentation tasks.

| | mIoU (↑) | | | | | | | | |
| | Open-vocabulary seg. | | | | | | Unsupervised seg. | | |
| | MaskCLIP + Ours | | | TCL + Ours | | | Ours | | |
| # masks | AD150 | CS171 | City19 | AD150 | CS171 | City19 | AD150 | CS171 | City19 |
|---|---|---|---|---|---|---|---|---|---|
| 10 | 15.7 | **21.5** | 24.2 | 17.2 | **22.3** | 21.8 | 30.9 | 30.0 | 34.0 |
| 20 | **16.0** | 21.1 | 25.9 | **17.6** | 22.1 | 23.3 | 32.9 | 30.4 | 36.3 |
| 30 | 15.9 | 20.7 | **26.5** | 17.4 | 21.8 | 23.4 | 33.1 | **30.5** | 37.1 |
| 40 | 15.6 | 20.4 | 26.4 | 17.3 | 21.6 | **23.5** | **33.4** | 30.3 | **37.3** |

Table 5: **Effects of extracting feature maps from each cross-attention layer** in $16 \times 16$ upward block. Evaluated on ADE150K. No significant differences in performance.

| | mIoU (↑) | | |
| | layer 1 | layer 2 | layer 3 |
|---|---|---|---|
| Unsupervised seg. | **33.1** | **33.1** | 32.8 |

names. Note that the purpose of this evaluation is not to achieve the state-of-the-art results in open-vocabulary semantic segmentation, but quantitatively evaluate the preciseness of our segmentation maps by combining them with the noisy feature maps produced by the baselines.

# D  HYPERPARAMETER ANALYSIS

In this section, we analyze how the selection of the hyperparameters in our framework affects the segmentation quality. In this analysis, unsupervised semantic segmentation is evaluated under our modified evaluation protocol.

**Number of masks per image.** We first ensure that our quantitative results are not sensitive to the number of masks generated per image within a reasonable range (Here we experiment between 10 to 40). As presented in Table 4, we observe only a marginal difference in mIoU for both unsupervised semantic segmentation and open-vocabulary semantic segmentation, confirming the fairness and robustness of our experiment setup.

**Selection of cross-attention layer to extract feature maps.** In Section 3.2, we extract feature maps from one of the cross-attention layers to build low-resolution segmentation maps. Our selection is based on the previous observation (Luo et al., 2023; Karazija et al., 2023) that layers at $16 \times 16$ blocks are most semantically meaningful. Note that there are three cross-attention layers at $16 \times 16$ resolution on the upward path. However, as reported in Table 5, we observed similar performance regardless of which $16 \times 16$ cross attention layer was selected.

**Timesteps for extracting feature maps.** We experiment with varying the timesteps of extracting feature maps. Recall that the extracted feature maps are used for generating low-resolution segmentation maps, therefore the feature maps are desired to clearly capture all the contents presented in the images. Intuitively, adding large noise (i.e., large timestep) ambiguates object layouts and should be avoided. Indeed, as visualized in Figure 12, when the timestep increases, our framework struggles to distinguish neighbouring objects in the images. Table 6 also highlights the performance degradation when adding larger noise.

**Modulation timestep and strength.** The influence of text captions on the generated images is known to vary throughout the denoising process (Balaji et al., 2022; Hertz et al., 2022; Patashnik et al., 2023). Given that our modulation alters the feature maps of cross-attention layers, where the text captions interact with the U-Net's feature maps, the influence of our modulation is also expected to be different throughout the timesteps. As visualized in Figure 13, when the timestep is too small, our modulation did not effectively affect the pixel values of the corresponding semantic objects and produced coarse segmentation masks. Conversely, when the timestep is too large, some segmen-

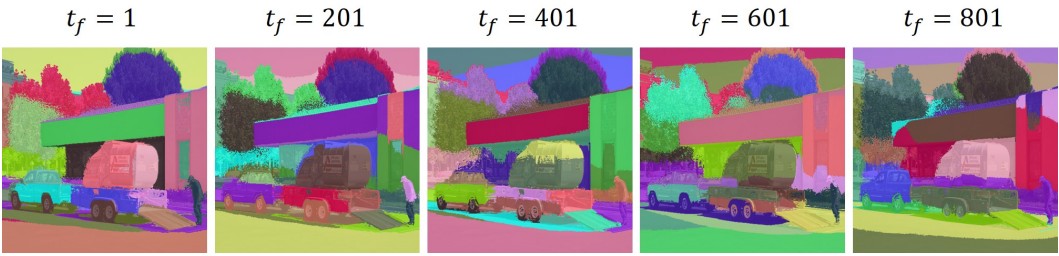

Figure 12: **Varying the timesteps of extracting feature maps.** As the timestep grows, the generated segmentation maps fail to distinguish neighboring objects.

Table 6: **Varying the timestep of extracting feature maps.** Evaluated on unsupervised semantic segmentation and open-vocabulary semantic segmentation (MaskCLIP + Ours) w/ ADE20K

| Timestep | mIoU (↑) | | | | |
| --- | --- | --- | --- | --- | --- |
| | 1 | 201 | 401 | 601 | 801 |
| Unsupervised seg. | **33.1** | 32.8 | 31.8 | 29.8 | 26.9 |
| Open-vocabulary seg. | **15.9** | 15.8 | 15.7 | 15.2 | 13.9 |

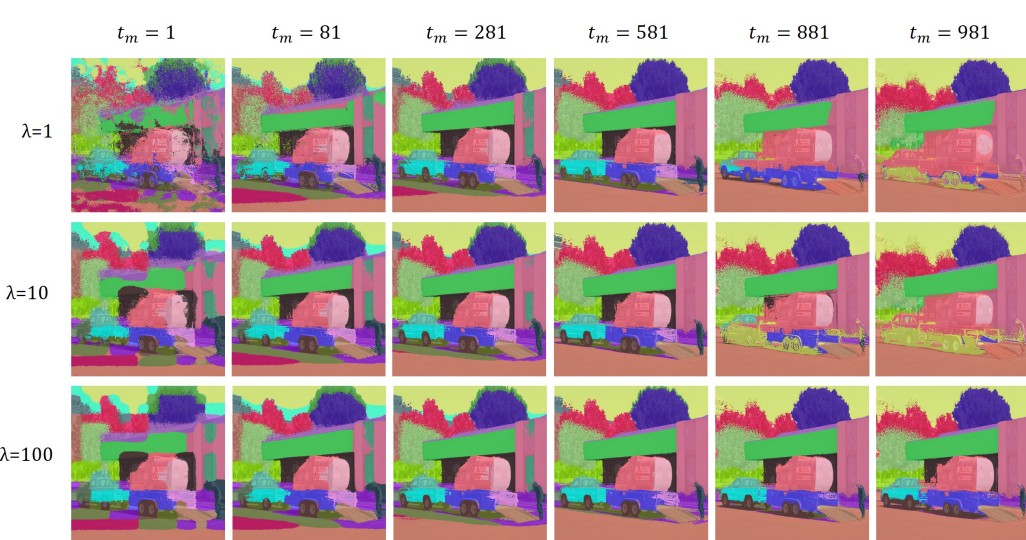

Figure 13: Effects of modulation timestep $t_m$ vs modulation strength $\lambda$.

tation masks were absorbed by others, likely because detailed semantic parts are not distinguished at large timesteps. The performance difference is also quantitatively verified in Table 7, where the early-middle timesteps achieve the best performance. Interestingly, our framework consistently generates visually decent segmentation maps when the modulation strength $\lambda$ is large enough. However, as reported in Table 8, slight degraded performance is observed when increasing $\lambda$ excessively. Nevertheless, we find that early-middle timesteps produce perceptually plausible segmentation masks, working effectively with both small and large $\lambda$.

**Selection of cross-attention layer to modulate.** There are three cross-attention layers in $16 \times 16$ upward modular blocks, where the first and the third ones are known to control generated contents and appearance respectively (Voynov et al., 2023). In our experiments, we modulate the third cross-attention layer by replacing its original computation $f\left(\sigma\left(\frac{QK^T}{\sqrt{d}}\right) \cdot V\right)$ with the modulated computation $f\left(\sigma\left(\frac{QK^T}{\sqrt{d}}\right) \cdot V\right) + cM$ (See 3.3 for more details). Here, we experiment with differ-

Table 7: **Varying the modulation timestep** $t_m$. Evaluated on unsupervised semantic segmentation w/ ADE20K

| | mIoU ($\uparrow$) | | | | | |
|---|---|---|---|---|---|---|
| $t_m$ | 1 | 81 | 281 | 481 | 681 | 881 |
| Unsupervised seg. | 29.0 | 31.8 | **33.1** | **33.1** | 31.5 | 27.4 |

Table 8: **Varying the modulation strength** $\lambda$. Evaluated on unsupervised semantic segmentation w/ ADE20K

| | mIoU ($\uparrow$) | | | |
|---|---|---|---|---|
| $\lambda$ | 1 | 10 | 100 | 1000 |
| Unsupervised seg. | 32.7 | **33.1** | 32.8 | 32.6 |

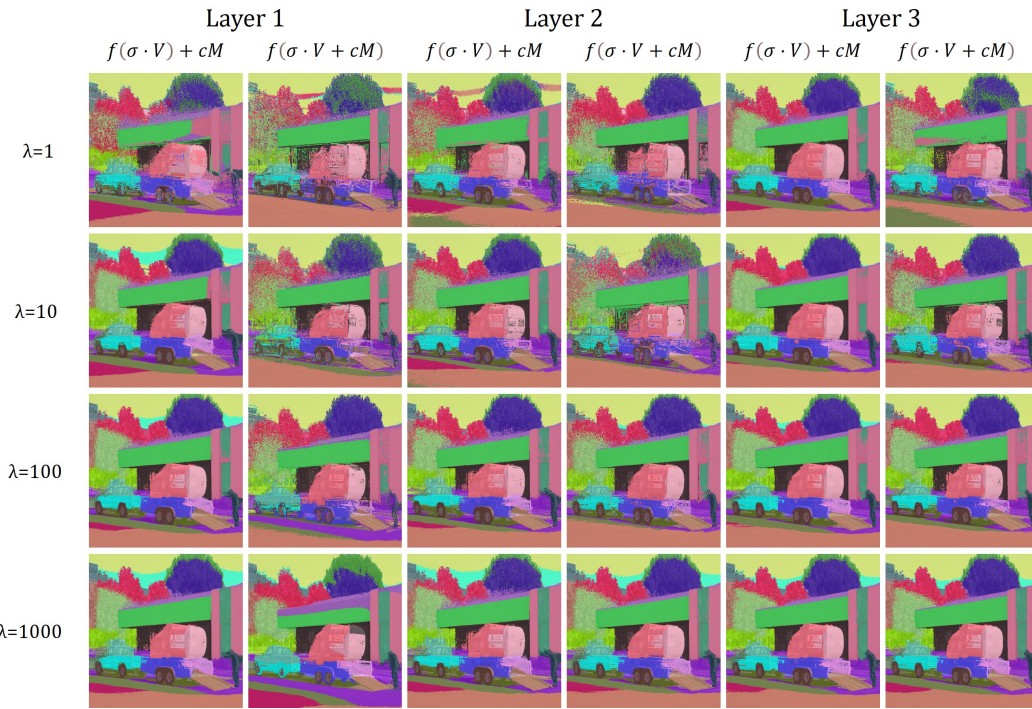

Figure 14: Effects of modulating different cross-attention layers vs different computation ($f\left(\sigma \cdot V\right) + cM$, $f\left(\sigma \cdot V + cM\right)$) vs different $\lambda$. For cross-attention layers, we experiment with the three different layers in $16 \times 16$ upward modular blocks. Note that we abbreviate $\sigma\left(\frac{QK^T}{\sqrt{d}}\right)$ to $\sigma$ for convenience.

ent modulated computation and cross-attention layers. Specifically, for modulated computation, we consider $f\left(\sigma\left(\frac{QK^T}{\sqrt{d}}\right) \cdot V + cM\right)$, which adds an offset $c$ before the fully connected layer $f$. As visualized in Figure 14, both computations produce visually plausible segmentation maps, indicating that our framework is not sensitive to the place of offset injection. For the choice of different cross attention layers, we observe a clear difference between the first layer and the third layer, likely because these layers are responsible for different attributes of the generated images and image pixels are affected differently. Overall, the third cross-attention layer produces relatively decent segmentation maps regardless of the modulated computations and $\lambda$. We conjecture that modulating the third layer naturally leads to changing the values of the pixels of semantic objects, which might be attributed to the fact that the third layer is controlling appearance.

Table 9: **Ablating attention injection.** Evaluated on unsupervised semantic segmentation and open-vocabulary semantic segmentation (MaskCLIP + Ours) w/ ADE20K

|  | mIoU (↑) | |
| --- | --- | --- |
|  | Unsupervised seg. | Open-vocabulary seg. |
| Ours w/o attention injection | 31.8 | 15.6 |
| Ours | **33.1** | **15.9** |

**Attention injection.** In Section 3.3, we fix the attention maps of all the attention layers to the original ones during the modulated denoising process. This is a technique used in image editing to preserve the structure of the original images. Figure 15 presents the comparison between our framework with and without the attention injection. While both produced visually plausible segmentation maps, our framework with attention injection preserves more detailed structures than the other (See person in the figure). The effectiveness is also quantitatively evident in Table 9.

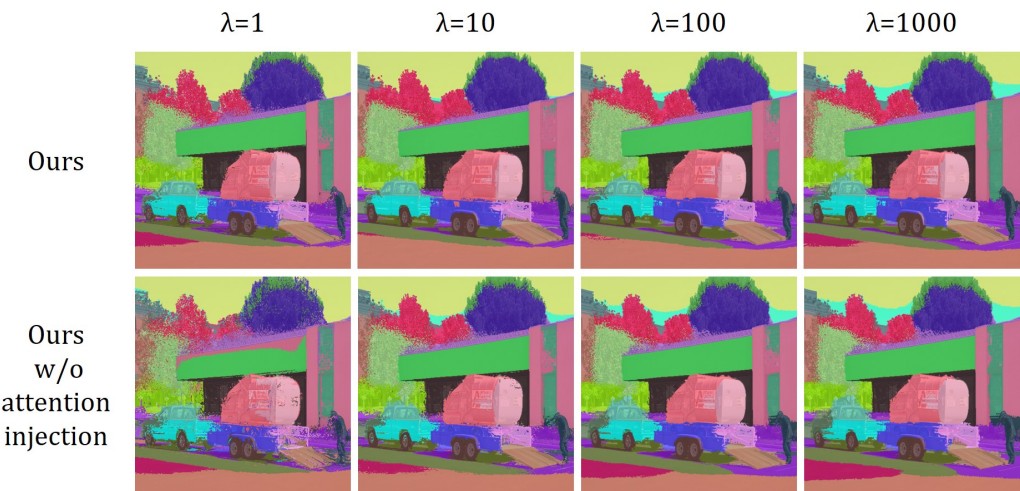

Figure 15: Ablating attention injection.

## E ADDITIONAL QUALITATIVE ANALYSIS

The additional qualitative results of ADE150K (Zhou et al., 2019), PASCAL-Context (Mottaghi et al., 2014) and COCO-Stuff (Caesar et al., 2018) are shown in Figure 16, 17, 18, respectively.

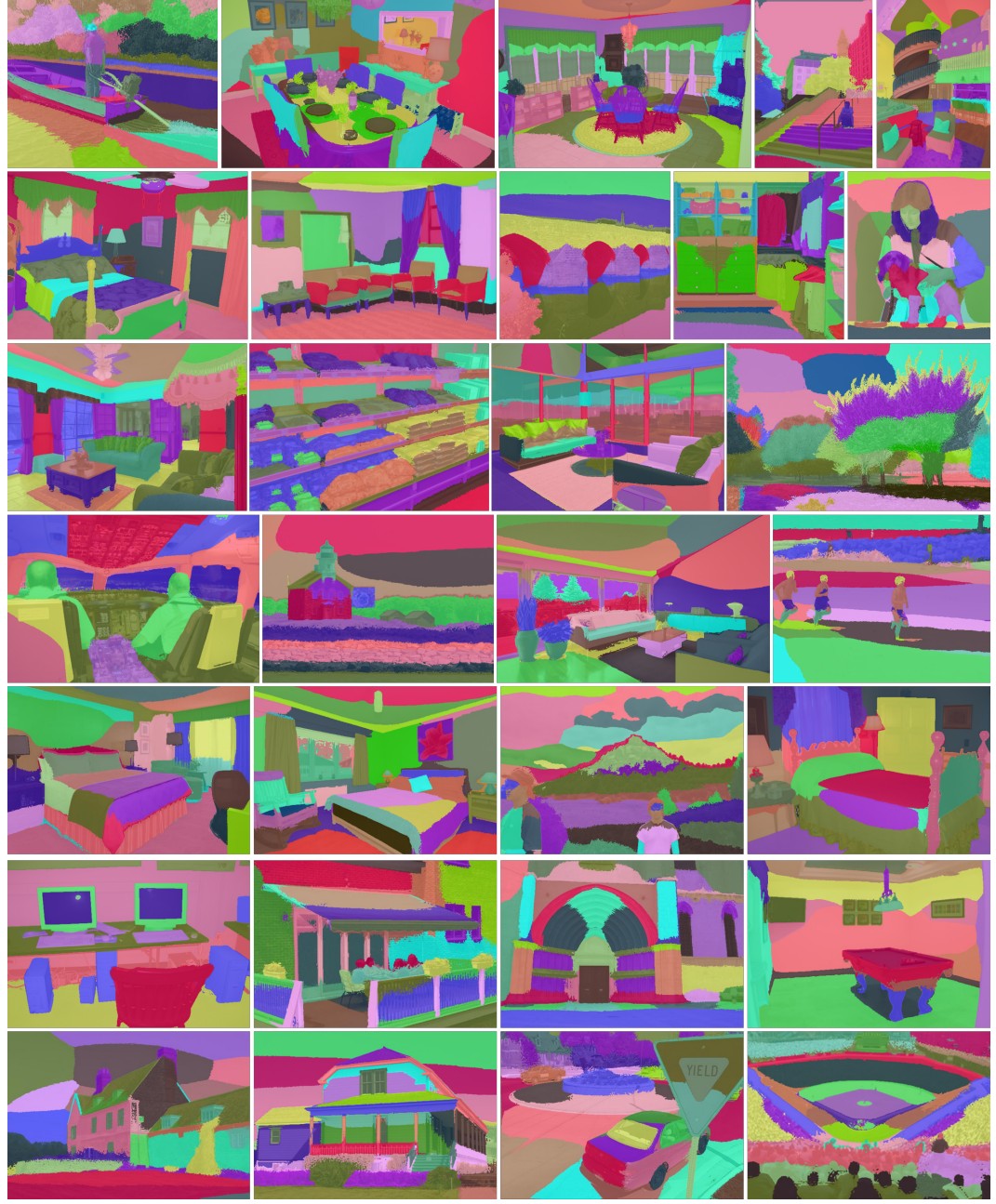

Figure 16: Examples of our produced segmentation maps on ADE20K (Zhou et al., 2019)

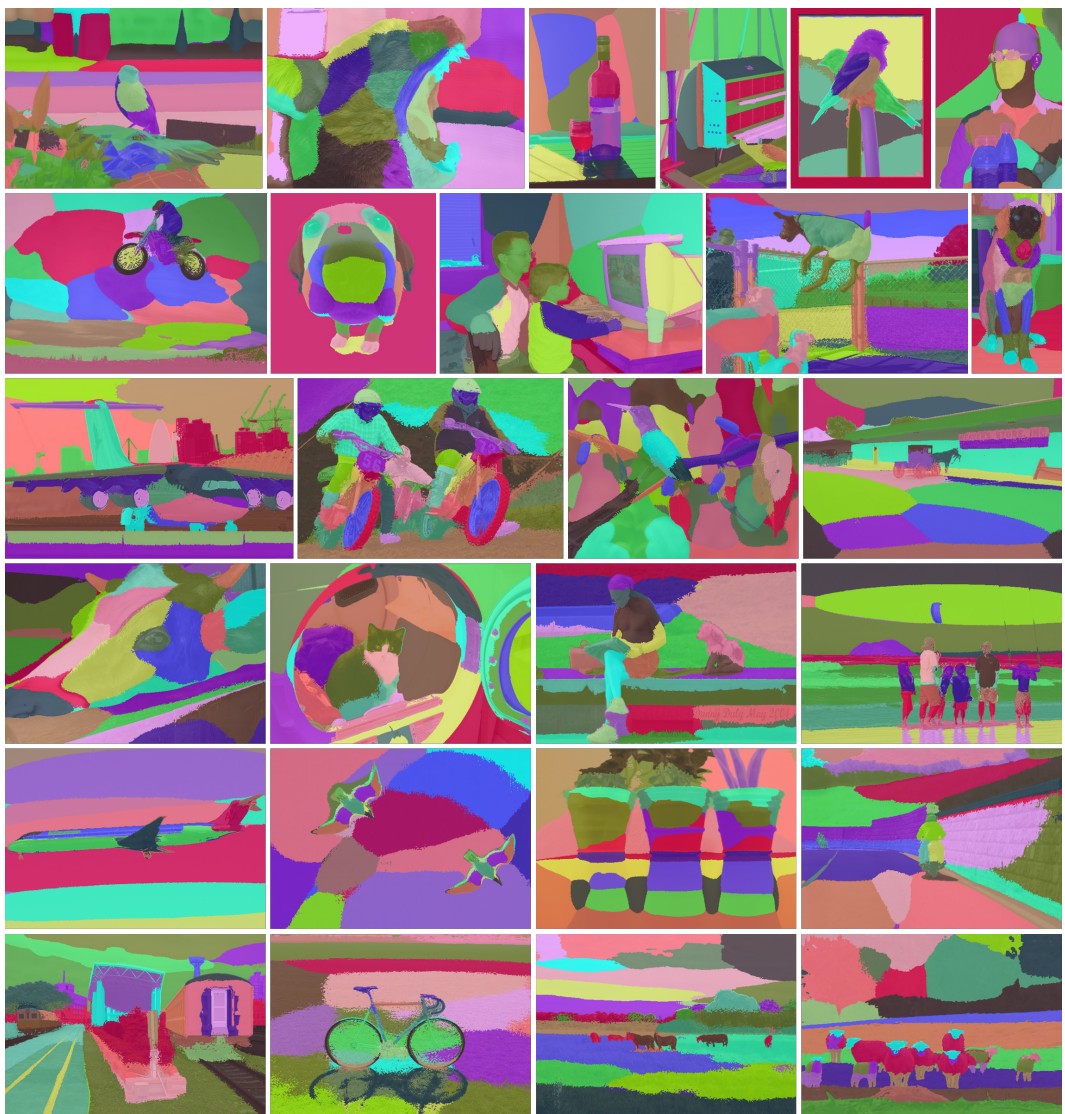

Figure 17: Examples of our produced segmentation maps on PASCAL-Context (Mottaghi et al., 2014)

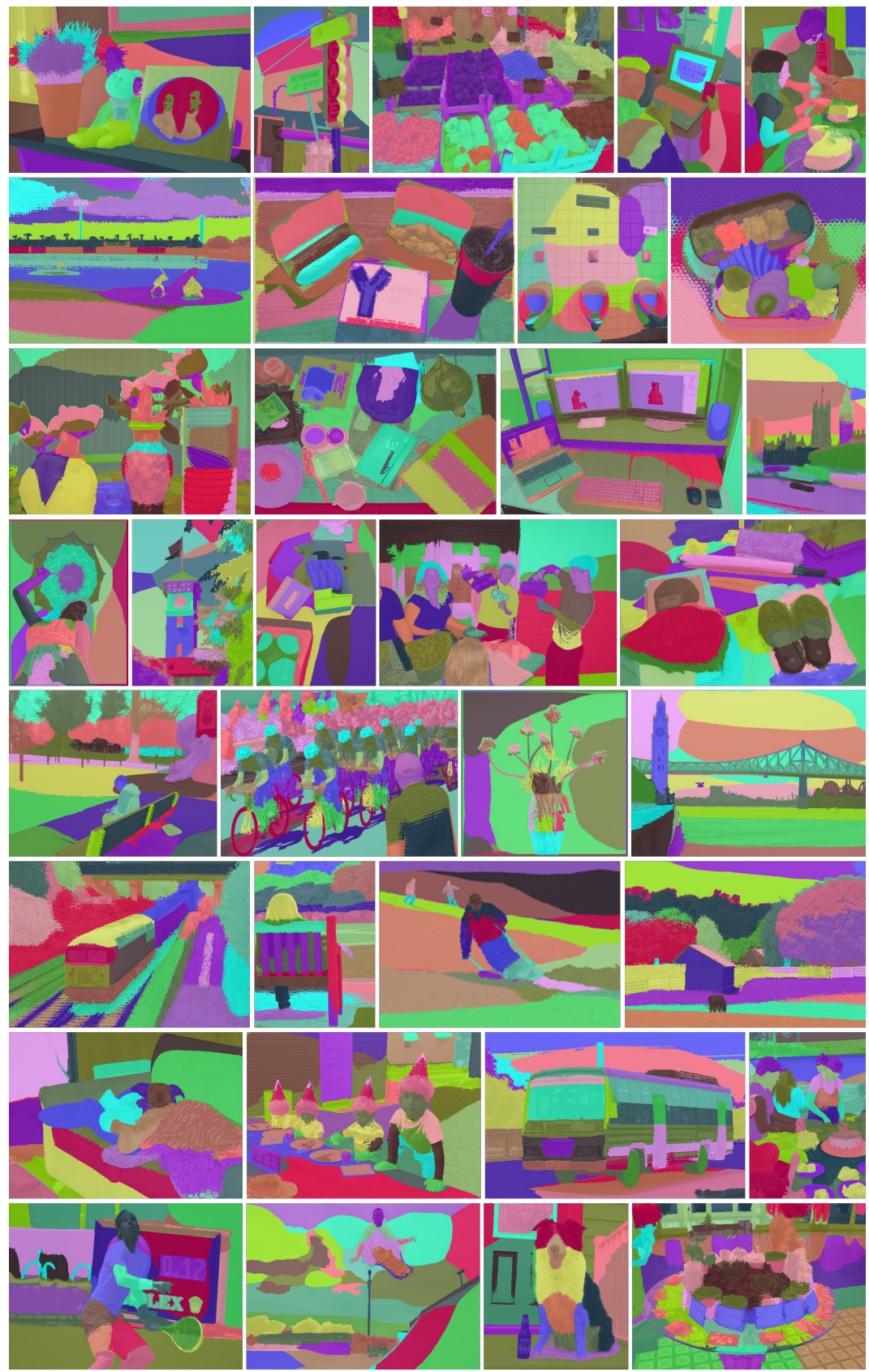

Figure 18: Examples of our produced segmentation maps on COCO-Stuff (Caesar et al., 2018)

