# OpenReview forum: "EmerDiff: Emerging Pixel-level Semantic Knowledge in Diffusion Models"
_ICLR.cc/2024/Conference — ICLR 2024 poster_

### Official Review · Reviewer_EgET · 2023-10-30

**Soundness:** 3 good
**Presentation:** 3 good
**Contribution:** 3 good
**Rating:** 6
**Confidence:** 1

**Summary:**

Diffusion models are being studied for their ability to perform semantic segmentation tasks. However, previous works relied on additional supervision to produce fine-grained segmentation maps. The authors of this study utilized semantic knowledge extracted from Stable Diffusion (SD) to build an image segmentor that can generate fine-grained segmentation maps without additional training. The study overcomes the challenge of identifying semantic correspondences between image pixels and spatial locations of low-dimensional feature maps by analyzing SD's generation process. The produced segmentation maps were well delineated and captured detailed parts of the images, indicating highly accurate pixel-level semantic knowledge in diffusion models.

**Strengths:**

A well-written paper with interesting ideas! Actually, I am not very familiar with this topic. So, I can only make some moderate comments.

**Weaknesses:**

1. While we would love to unlock the perception of SD, I find this article's approach to be inelegant and not performing well.
2. Some of the existing solutions also have no comparison at all (such as INSTRUCTCV).

**Questions:**

See weakness.

---

> ### Author Response · Authors · 2023-11-14
> **Thank you for feedback**
>
> Dear Reviewer EgET,
>
> We thank you for your engagement with our submission.
>
> Below we address your concerns:
>
> > While we would love to unlock the perception of SD, I find this article's approach to be inelegant and not performing well.
>
> > Some of the existing solutions also have no comparison at all (such as INSTRUCTCV).
>
> One of the purposes of building our image segmentor is to visualize pixel-level semantic understanding of pre-trained diffusion models. In the previous works, segmentation with diffusion models was typically achieved by incorporating additional training on segmentation datasets, which, however, blurred how much pre-trained diffusion models alone understand pixel-level semantic relations. Our image segmentor, which is solely built on the pre-trained diffusion models without any additional training, has revealed that the pre-trained diffusion models already possess highly accurate pixel-level semantic knowledge, which provides an important insight into the current research trends of "unlocking the perception of" generative models.
>
> Following that, we did not include the comparison with the existing methods that require additional supervision with segmentation datasets, including DatasetDDPM [1], ODISE [2], and INSTRUCTCV. We kindly remind that INSTRUCTCV is a concurrent work released on arXiv after the ICLR submission deadline, and therefore we were not able to cite it in our paper. We are planning to mention it in the related work section.
>
> We are happy to address any other questions.
>
> [1] Dmitry Baranchuk, Andrey Voynov, Ivan Rubachev, Valentin Khrulkov, and Artem Babenko. Labelefficient semantic segmentation with diffusion models. In International Conference on Learning Representations, 2022.
>
> [2] Jiarui Xu, Sifei Liu, Arash Vahdat, Wonmin Byeon, Xiaolong Wang, and Shalini De Mello. Openvocabulary panoptic segmentation with text-to-image diffusion models. In Proceedings of the IEEE/CVF Conference on Computer Vision and Pattern Recognition, pp. 2955–2966, 2023.

---

### Official Review · Reviewer_CUiU · 2023-10-30

**Soundness:** 3 good
**Presentation:** 3 good
**Contribution:** 3 good
**Rating:** 6
**Confidence:** 5

**Summary:**

This paper presents an approach to leverage diffusion models to produce image segmentation maps without additional supervision. The authors first cluster low-resolution feature map which is believed to contain semantic information into a set of masks. To produce high-resolution segmentation, the authors find a region in the image space that is highly correlated to each mask of the set. The process is called modulated denoising process. The experimental results are comprehensive to demonstrate the ability of the proposed method to produce segmentation maps of given images.

**Strengths:**

1. The paper introduces an approach for generating segmentation maps by leveraging semantic knowledge extracted from diffusion models. This approach does not require additional annotations.
2. The proposed framework is extensively evaluated on multiple scene-centric datasets, including COCO-Stuff, PASCAL-Context, ADE20K, and Cityscapes. The qualitative and quantitative results demonstrate the method's effectiveness in producing segmentation maps.
3. The paper provides qualitative examples of failure cases.
4. The paper is well-written.

**Weaknesses:**

1. This approach using many sensitive hyper-parameters, which is need to be tuned very carefully for each dataset. It is not easy for large dataset and we don't have a validation set.
2. Although the paper shows the ablation study for choosing some hyper-parameters, it stills lacks some others such as the level of cross-attention layer for low-resolution feature map, modulating timestep (should have quantitative numbers)
3. The resulting segmentation seems to be over-segmented (Fig. 1), sometimes just like superpixels, still need semantic segmentation labels to align and group these over-segmentation regions to have meaningful masks, it is different from SAM.
4. The modulated denoising process seems to be slow since it has to go through all steps of UNet and decoder of VAE to get the corresponding mask in high-resolution image for each mask.

**Questions:**

1. Can the authors provide comparisons of the runtime between different methods in Table 2?
2. Is it possible to use this method for synthesizing semantic segmentation datasets to train a semantic segmenter?

---

> ### Author Response · Authors · 2023-11-13
> **Thank you for questions and feedback**
>
> Dear Reviewer CUiU,
>
> We thank you for your valuable feedback.
>
> Below we address your specific concerns and questions:
>
> > This approach using many sensitive hyper-parameters, which is need to be tuned very carefully for each dataset. It is not easy for large dataset and we don't have a validation set.
>
> We kindly remind that we use the **same hyper-parameters for all the datasets** (COCO-Stuff, PASCAL-Context, ADE20K, and Cityscapes). Although laborious hyperparameter tuning has been the common issue in the literature of unsupervised semantic segmentation (e.g. STEGO [1]), we avoid this problem by utilizing the powerful generative model (i.e. Stable Diffusion) trained on images of diverse domains.
>
> > Although the paper shows the ablation study for choosing some hyper-parameters, it stills lacks some others such as the level of cross-attention layer for low-resolution feature map, modulating timestep (should have quantitative numbers)
>
> We appreciate your suggestion. We will address this in the followup response.
>
> > The resulting segmentation seems to be over-segmented (Fig. 1), sometimes just like superpixels
>
> In Figure 1, we deliberately set the number of masks high (e.g. 30) for visualization purposes. As demonstrated later in Figure 5, if we keep the number of segmentation masks low, we can easily confirm the semantic awareness of the obtained segmentation maps, which is different from superpixels.
>
> > still need semantic segmentation labels to align and group these over-segmentation regions to have meaningful masks
>
> As an example to contextualize our obtained segmentation masks without any segmentation labels, we have experimented with the task of annotation-free open-vocabulary semantic segmentation (Please see Figure 7), where we use the off-the-shelf image encoder (e.g. CLIP) trained on image-text pairs to produce text-aligned embeddings for each mask. By associating the produced masks with the text embeddings, we overcome the issue of over-segmentation.
>
> > it is different from SAM.
>
> One of the goals of this study is to build an image segmentor that can visualize semantic understanding of diffusion models at pixel level. By quantitatively evaluating our image segmentor in the tasks of unsupervised semantic segmentation and annotation-free open-vocabulary semantic segmentation, we demonstrate the existence of highly accurate pixel-level semantic knowledge in the diffusion models, which provides an important insight into the current research trends of utilizing internal representations of diffusion models for discriminative tasks [2][3][4]. We believe our image segmentor is valuable in the research community, although SAM, which is trained on massive segmentation datasets, could be more practically useful.
>
> > The modulated denoising process seems to be slow since it has to go through all steps of UNet and decoder of VAE to get the corresponding mask in high-resolution image for each mask.
>
> Please refer to the response to Reviewer D7CR within which we discuss the matter of speed of the method.
>
> > Can the authors provide comparisons of the runtime between different methods in Table 2?
>
> We appreciate your suggestion. We will address this in the followup response.
>
> > Is it possible to use this method for synthesizing semantic segmentation datasets to train a semantic segmenter?
>
> Since our work established the technique to accurately find the semantic correspondences between the sub-regions of low-resolution feature maps and the image pixels (Figure 3), it can be incorporated into the existing studies to enhance the quality of synthesized segmentation masks.
> For instance, previous works [5] [6] [7] relied on low-resolution feature maps to delimit the semantic objects from the input image, therefore they had to utilize some boundary refinement techniques to obtain fine-grained segmentation masks. Our method can replace this process by simply finding the semantic correspondences between the image pixels and the delimited region of the low-resolution feature maps. We think this is an interesting research direction which we leave to future work.

---

> > ### Author Response · Authors · 2023-11-13
> > **Thank you for questions and feedback (References)**
> >
> > References:
> >
> > [1] Mark Hamilton, Zhoutong Zhang, Bharath Hariharan, Noah Snavely, and William T. Freeman. Unsupervised semantic segmentation by distilling feature correspondences. In International Conference on Learning Representations, 2022.
> >
> > [2] Dmitry Baranchuk, Andrey Voynov, Ivan Rubachev, Valentin Khrulkov, and Artem Babenko. Labelefficient semantic segmentation with diffusion models. In International Conference on Learning Representations, 2022.
> >
> > [3] Jiarui Xu, Sifei Liu, Arash Vahdat, Wonmin Byeon, Xiaolong Wang, and Shalini De Mello. Openvocabulary panoptic segmentation with text-to-image diffusion models. In Proceedings of the IEEE/CVF Conference on Computer Vision and Pattern Recognition, pp. 2955–2966, 2023.
> >
> > [4] Daiqing Li, Huan Ling, Amlan Kar, David Acuna, Seung Wook Kim, Karsten Kreis, Antonio Torralba, and Sanja Fidler. Dreamteacher: Pretraining image backbones with deep generative models, 2023.
> >
> > [5] Weijia Wu, Yuzhong Zhao, Mike Zheng Shou, Hong Zhou, and Chunhua Shen. Diffumask: Synthesizing images with pixel-level annotations for semantic segmentation using diffusion models, 2023.
> >
> > [6] Chaofan Ma, Yuhuan Yang, Chen Ju, Fei Zhang, Jinxiang Liu, Yu Wang, Ya Zhang, and Yanfeng
> > Wang. Diffusionseg: Adapting diffusion towards unsupervised object discovery. arXiv preprint
> > arXiv:2303.09813, 2023.
> >
> > [7] Laurynas Karazija, Iro Laina, Andrea Vedaldi, and Christian Rupprecht. Diffusion Models for ZeroShot Open-Vocabulary Segmentation. arXiv preprint, 2023.

---

> ### Author Response · Authors · 2023-11-20
> **Thank you for questions and feedback (Follow up)**
>
> Dear Reviewer CUiU,
>
> We thank you again for your valuable feedback and suggestions.
>
> Here we address the remaining questions.
>
> > Although the paper shows the ablation study for choosing some hyper-parameters, it stills lacks some others such as the level of cross-attention layer for low-resolution feature map, modulating timestep (should have quantitative numbers)
>
> We appreciate your suggestions. We have included the additional results in Appendix D (highlighted in red).
>
> Regarding the selection of cross-attention layers to build low-resolution segmentation maps, our choice of $16 \times 16$ resolution is based on the previous observation (e.g. Figure7 of [8])  that the layers at $16 \times 16$ resolution are most semantically informative. Since there are three cross-attention layers at $16 \times 16$, we additionally experimented with the effects of varying the choice of these three layers. It turns out that the choice amoung these three layers does not differentiate the performance (mIOU: 33.1, 33.1, 32.8, respectively), which is also reported in Table 5.
>
> For the modulated timesteps, we have added the quantitative analysis in Table 7.  Our results are consistent with the qualitative results, and modulating at an early-middle timestep achieves the best performance. We note that there is some flexibility in choosing the modulation timesteps $t_m$. For example, in our experiments, both $t_m = 281$ and $t_m = 481$ attain the same best performance.
>
> > Can the authors provide comparisons of the runtime between different methods in Table 2?
>
> Due to time constraints, we were not able to provide the runtime comparison in Table 2. However, we have included the discussion about the runtime in Section 4.1 (highlighted in blue). We hope that this addresses your concerns about the runtime of our method. Admittedly, the runtime is a disadvantage of diffusion-based methods, but we expect this to be significantly faster in the future as there is a lot of active research in speeding up diffusion models.
>
> We hope that we have addressed all your concerns and questions.
> If you have any remaining questions, we would happily continue to engage in discussions. Otherwise, we kindly ask you to consider increasing your rating as we believe our method gives new insight into the area of uncovering the semantic understanding of the diffusion models.
>
> Thank you,
>
> The authors
>
> References:
>
> [8] Grace Luo, Lisa Dunlap, Dong Huk Park, Aleksander Holynski, and Trevor Darrell. Diffusion hyperfeatures: Searching through time and space for semantic correspondence. arXiv, 2023

---

> ### Comment · Reviewer_CUiU · 2023-11-22
>
> I would like to thank the authors for their response. However, my concerns regarding the following points have not been addressed successfully:
> 1. About hyperparam tuning. Have the authors try to tune differently on different datasets. If the results get better, it will show that each dataset needs to be carefully tuned.
> 2. About over-segmentation. Again, you need to tune the number of clusters in K-means.
> 3. About SAM. SAM is considered a foundation model (trained on large-scale datasets and promising to cover almost daily-life photos as tested in this paper). Why don't we just use it?
> 4. About running time. Again, not address my concern since it is still slow compared to other approaches.
>
> Therefore, I keep my original rating. While the paper's concept is strong, its practicality appears limited, especially when established alternatives like SAM exist.

---

> ### Author Response · Authors · 2023-11-22
> **Thank you for your response**
>
> Dear Reviewer CUiU,
>
> We sincerely thank you for your response to our rebuttal.
>
> > 3. About SAM [..] Why don't we just use it?
>
> > 4. About running time. [..]
>
> We understand your focus is on the practical side of our method. We would like to re-emphasize that the purpose of our study is to demonstrate the pixel-level semantic understanding of diffusion models, which we accomplished by inventing an unsupervised segmentation algorithm that solely relies on the pre-trained diffusion model. We believe this is an interesting research topic, because, intuitively, generative models should be able to capture the semantics of their generated images at pixel levels to synthesize images at high quality, but none of the previous works could extract such pixel-level semantic knowledge from the generative models in an unsupervised manner. Therefore, although our method is less practically useful compared to the supervised baselines, relying on SAM or other previous supervised approaches is not helpful in achieving our research goal.
>
> > 2. About over-segmentation. [..]
>
> We kindly remind you that, as demonstrated in Table 4, our experiments are not sensitive to the number of masks (K=10-40).
>
> >1. About hyperparam tuning. [..]
>
> We kindly remind you that, in our experiments, we achieved the performance that outperforms discriminative counterparts (e.g. DINO) in all the datasets with the unified hyperparameters. Since the purpose of our experiments is to verify a good pixel-level semantic understanding of diffusion models, applying careful hyperparameter tuning for each dataset deviates from our original research goal. We agree that investigating whether careful hyperparameter tuning yields better results is an interesting research question, which we leave for future works.
>
> We hope this addresses your concerns. We are happy to continue to discuss this with you until the last minitue of the discussion period!
>
> Thank you

---

### Official Review · Reviewer_aY52 · 2023-10-31

**Soundness:** 3 good
**Presentation:** 3 good
**Contribution:** 3 good
**Rating:** 6
**Confidence:** 4

**Summary:**

The paper introduces a new method to extract semantic segmentation maps for images with the help of the pre-trained Stable Diffusion model. Given an image, the proposed approach embeds it into a time step $x_t$, passes to the respective UNet, collects features at an upward 16x16 layer, and performs k-means clustering. To obtain the final segmentation at resolution 512x512, the method studies how much the final pixel values depend on different masked 16x16 feature areas. In effect, the proposed approach can provide pixel-level k-class segmentation maps for each provided image, without requiring any training or text prompts. The experiments demonstrate that the proposed method achieves good performance on the tasks of unsupervised semantic segmentation and annotation-free open-vocabulary semantic segmentation.

**Strengths:**

- The paper finds good use of powerful T2I diffusion models for segmentation tasks. While there exist previous methods extracting segmentation masks via latent spaces of DMs, applying them to the class of latent diffusions was not obvious due to the mismatch in resolutions. The paper proposes an adequate solution.
- The paper demonstrates impressive results. The proposed method segments images into K classes, and the pixels belonging to the same objects tend to fall within same segments.
- The method does not require re-training the DM, test-time optimisation, or text prompts. It can be applied to images of diverse datasets rather straightforwardly.
- The method shows impressive results when applied to downstream applications like unsupervised segmentation.

**Weaknesses:**

- The proposed method is not too novel in terms of the main idea. Transforming internal UNet feature representations into segmentation maps via K-means was explored in prior work (e.g., Baranchuk et al, Fig 4), although perhaps not in the context of Stable Diffusion. The modification to make it work with SD (Sec. 3.3) makes sense and works, but is not particularly educating or insightful, to my opinion.
- Overall, the inventive step of the method is strongly tailored for the family of Stable Diffusion models. Although this makes sense as many state-of-the-art DMs belong to this family, the method still may become not so much applicable in case of other types of DMs.
- [Xu et al, CVPR23] noted that using pre-trained SD with zero text input is suboptimal for feature extraction. Did the authors observe any suboptimalities in using zero text line as inputs, or did they experiment with providing textual descriptions to the model?

**Questions:**

I would appreciate the authors' comment on the points from the weaknesses section.

---

> ### Author Response · Authors · 2023-11-12
> **Thank you for questions and feedback**
>
> Dear Reviewer aY52,
>
> We thank you for your thoughtful feedback.
>
> Below we will address your specific concerns and questions:
>
>
> > The proposed method is not too novel in terms of the main idea [...] not particularly educating or insightful, to my opinion.
>
> Our method is indeed motivated by the previous observation that applying k-means on specific intermediate feature representations yield semantically meaningful clusters (as mentioned in the second paragraph of the introduction section). However, for many classes of generative models (LDMs, DMs, GANs), these semantically useful features are spatially low dimensional. For instance, Figure 4 of Baranchuk et al [1] suggests only features with a resolution of $64 \times 64$ or lower in guided diffusion (DM) yield semantically interpretable clusters, whereas the image resolution of the model is $256 \times 256$. This phenomenon is also observed in StyleGAN (Figure 9-12 in the appendix of [2]). Therefore, the problem of "mismatch in resolutions" is not specific to LDMs, and extracting pixel-level semantic relations from such generative models is a non-trivial task. To the best of our knowledge, we are the first to tackle this problem with Stable Diffusion.
>
> As you have pointed out, our primary contribution is Section 3.3, where we show that the semantic knowledge formed in the spatially lower-dimensional layers can be accurately rendered onto the image pixel space by realizing the correspondences between the low-dimensional feature space and image pixels. Despite the simplicity of our idea, it significantly facilitates the visualization and interpretation of lower-dimensional feature maps (Please see Figure 4 of our work). Our work utilizes this technique to demonstrate that the lower-dimensional feature representations of SD are strongly tied with the semantic objects in the image, which provides an important insight to the research community that utilizes internal representations of diffusion models for discriminative tasks [1][3][4].
>
> > Overall, the inventive step of the method is strongly tailored for the family of Stable Diffusion models [...]
>
> Semantically meaningful feature representations are commonly observed inductive biases in many generative models, including LDMs, DMs, and GANs [2]. However, in all cases, such feature representations are of lower resolutions than the image resolution.
>
> Although we leave experimenting with other types of generative models for future work, our core idea of measuring "how much the final pixel values depend on different masked feature areas" by perturbing the values of the sub-regions of the feature maps should technically apply to all these models. Specifically, we believe our pipeline does not contain any parts that are "strongly" tied to the model being a latent diffusion model. We are curious to know which part you believe to be strongly tailored for the family of latent diffusions.
>
> > [Xu et al, CVPR23] noted that using pre-trained SD with zero text input is suboptimal for feature extraction [...]
>
> For this project, our aim was to extract pixel-level semantic knowledge without relying on any additional training or supervision. As such, we didn't provide any textual descriptions in our experiments, and we found that even without captions it yields good results. This finding is in line with Xu et al [3] (Table 4), where they achieved similarly strong performance without injecting meaningful captions.
> Combining this method with an automatic image-captioning model could be an interesting direction which we leave to future work.
>
> References:
>
> [1] Dmitry Baranchuk, Andrey Voynov, Ivan Rubachev, Valentin Khrulkov, and Artem Babenko. Labelefficient semantic segmentation with diffusion models. In International Conference on Learning
> Representations, 2022.
>
> [2] Edo Collins, Raja Bala, Bob Price, and Sabine Susstrunk. Editing in style: Uncovering the local
> semantics of gans. In Proceedings of the IEEE/CVF Conference on Computer Vision and Pattern
> Recognition, pp. 5771–5780, 2020.
>
> [3] Jiarui Xu, Sifei Liu, Arash Vahdat, Wonmin Byeon, Xiaolong Wang, and Shalini De Mello. Openvocabulary panoptic segmentation with text-to-image diffusion models. In Proceedings of the
> IEEE/CVF Conference on Computer Vision and Pattern Recognition, pp. 2955–2966, 2023.
>
> [4] Daiqing Li, Huan Ling, Amlan Kar, David Acuna, Seung Wook Kim, Karsten Kreis, Antonio Torralba, and Sanja Fidler. Dreamteacher: Pretraining image backbones with deep generative models,
> 2023.

---

> ### Author Response · Authors · 2023-11-20
> **Follow up**
>
> Dear Reviewer aY52,
>
> We thank you again for your thoughtful feedback, especially in the weakness section.
> We hope that we have addressed all your concerns and questions.
>
> If you have any other concerns, we are more than happy to discuss them with you. If there are no remaining concerns, we kindly ask you to consider raising your rating.
>
> Thank you,
>
> The authors

---

> ### Comment · Reviewer_aY52 · 2023-11-21
> **Thanks for getting back**
>
> I thank the authors for their reply and their explanations.
>
> After reading the response, I remain with my opinion that the issue of "mismatch in resolutions" is more specific to LDMs rather than to other types of models. For example, in GANs, this problem is usually bypassed by stacking features at different layers (and resolutions), e.g. [Tritrong et al, CVPR21]. Even if the main semantics is contained in mid-size features, higher-resolution features can contribute to refine the overall prediction. Similar approach can be observed for DDPM-style DMs, e.g., [Baranchuk et al]. I agree with the authors that this solution does not work for LDMs because they do not have full-resolution feature maps. For this type of models the proposed solution indeed solves an existing problem. It is not convincing to me that this solution is much relevant in other contexts. So to answer the question of the authors, I would say I perceive the *motivation* of the method a lot tailored for LDMs.
>
> I do still acknowledge the quality of results and potential impact of the paper, which makes me remain with my positive score.

---

> ### Author Response · Authors · 2023-11-21
> **Thank you for your response**
>
> Dear Reviewer aY52,
>
> We sincerely thank you for your response to our rebuttal and your acknowledgment of the potential impact of our paper.
>
> We initially tried applying k-means on the stacked features at different layers of StyleGAN, however, we found getting fine-grained segmentation masks with such a heuristic approach is challenging (Please also see Figure 7 of [5], where their bedroom and church results are noisy and less semantically aware due to the involvement of noisy feature maps at higher layers). We kindly remind that both [Tritrong et al, CVPR21] and [Baranchuk et al] were able to produce edge-aligned segmentation masks because of its reliance on additional training on segmentation datasets, while our method does not require such additional datasets/training.
>
> We hope this addresses your concern.  We are happy to continue to discuss this with you!
>
> References:
>
> [5] Xu, Jianjin, Zhaoxiang Zhang, and Xiaolin Hu. "Extracting Semantic Knowledge from GANs with Unsupervised Learning." IEEE Transactions on Pattern Analysis and Machine Intelligence (2023).

---

### Official Review · Reviewer_D7CR · 2023-11-02

**Soundness:** 3 good
**Presentation:** 3 good
**Contribution:** 3 good
**Rating:** 6
**Confidence:** 4

**Summary:**

This paper leverages pre-trained stable diffusion to achieve unsupervised semantic segmentation and annotation-free open-vocabulary semantic segmentation. The core idea is based on the observation that changing the values of a sub-region of low-resolution feature maps triggers the notable value changes of values in semantically related regions of the generated images. The  overall idea is interesting and reasonable.

**Strengths:**

- The proposed idea of exploring semantic knowledge extracted from Stable Diffusion (SD) and building an image segmentor is novel and interesting.
- The finding about the changes in sub-region of low-resolution feature maps and significant changes of semantically related pixels in generated images of stable diffusion model is interesting.
- The experimental results are convincing.

**Weaknesses:**

- Since one needs to change the values of every sub-region of low-resolution feature maps and examine the changes in the corresponding generated images, the whole process may take much time.

**Questions:**

Is there a more elegant way to establish the correspondence between the semantically related pixels and the sub-region in low-resolution feature maps. The current way of changing the the values in every sub-region of low-resolution feature maps and examining the changes in generated images works, but requires much time (I think).

---

> ### Author Response · Authors · 2023-11-12
> **Thank you for questions and feedback**
>
> Dear Reviewer D7CR,
>
> We thank you for your constructive feedback.
>
> Below we address your concerns about the runtime:
>
> > Since one needs to change the values of every sub-region of low-resolution feature maps [...] the whole process may take much time.
>
> > Is there a more elegant way to establish the correspondence [...], but requires much time (I think).
>
> Indeed, our method requires running the independent modulated denoising process for each mask, which may take much time with the naive implementation. A straightforward optimization would be parallelizing the modulated denoising processes, which is effective as the denoising processes do not involve backpropagation. Additionally, the modulated denoising processes are only required to start from the timestep to apply modulation (i.e. modulated timestep $t_{m}$). Since our paper sets $t_{m} = 281$ (out of $T=1000$) under DDPM sampling scheme of $50$ steps, it only requires up to $15$ steps to run each modulated denoising process. However, we agree that our method is still computationally expensive and would love to see whether further optimization is possible, which we leave for future work.
>
> As you have pointed out, our primary focus was to present the key finding that "changing the values of a sub-region of low-resolution feature maps triggers the notable value changes of values in semantically related regions of the generated images", which enables better visualization and interpretation of semantically meaningful low-dimensional feature maps (Please see Figure 4 (or  Figure 1 of [2]) for a comparison against the naive upsampling approach).  We utilize this technique to demonstrate that the low-dimensional feature representations are highly accurately tied with the semantic objects in the image, which provides an important insight into the current research trends of utilizing internal representations of diffusion models for discriminative tasks [1][2][3].
>
> References:
>
> [1] Dmitry Baranchuk, Andrey Voynov, Ivan Rubachev, Valentin Khrulkov, and Artem Babenko. Labelefficient semantic segmentation with diffusion models. In International Conference on Learning
> Representations, 2022.
>
> [2] Jiarui Xu, Sifei Liu, Arash Vahdat, Wonmin Byeon, Xiaolong Wang, and Shalini De Mello. Openvocabulary panoptic segmentation with text-to-image diffusion models. In Proceedings of the
> IEEE/CVF Conference on Computer Vision and Pattern Recognition, pp. 2955–2966, 2023.
>
> [3] Daiqing Li, Huan Ling, Amlan Kar, David Acuna, Seung Wook Kim, Karsten Kreis, Antonio Torralba, and Sanja Fidler. Dreamteacher: Pretraining image backbones with deep generative models,
> 2023.

---

> ### Author Response · Authors · 2023-11-20
> **Follow up**
>
> Dear Reviewer D7CR,
>
> We thank you again for your insightful feedback.
> We have incorporated the discussion about running time in our revised paper (in section 4.1, highlighted in blue).
>
> We are happy to address any other concerns you may have. If you found our response to be satisfactory, we kindly ask you to consider raising your rating.
>
> Thank you,
>
> The authors

---

### Meta-Review · Area_Chair_foSA · 2023-12-08

**Metareview:**

This paper proposes a novel method to obtain training-free open-vocabulary semantic segmentation by leveraging a pre-trained stable diffusion model. To this end the method performs k-means clustering in low-res feature maps of the denoising Unet, applied to a noised version of the input. The reviews appreciate the contributions and results of the paper, the rebuttal partially addressed the points raised in the rebuttal. The reviews unanimously recommend acceptance.

**Justification For Why Not Higher Score:**

The reviewers unanimously rate the paper as marginally above the threshold, which warrants acceptance, but not seem not seem to warrant spotlight or oral presentation.

**Justification For Why Not Lower Score:**

None of the reviewers advocates rejecting the paper

---

### Decision · Program_Chairs · 2024-01-16

Accept (poster)